# 5-Fluorouracil Encapsulated Chitosan-Cellulose Fiber Bionanocomposites: Synthesis, Characterization and In Vitro Analysis towards Colorectal Cancer Cells

**DOI:** 10.3390/nano11071691

**Published:** 2021-06-28

**Authors:** Mostafa Yusefi, Hui-Yin Chan, Sin-Yeang Teow, Pooneh Kia, Michiele Lee-Kiun Soon, Nor Azwadi Bin Che Sidik, Kamyar Shameli

**Affiliations:** 1Malaysia-Japan International Institute of Technology, Universiti Teknologi Malaysia, Jalan Sultan Yahya Petra, Kuala Lumpur 54100, Malaysia; myusefi175@gmail.com; 2Department of Medical Sciences, School of Medical and Life Sciences, Sunway University, Jalan Universiti, Bandar Sunway, Selangor Darul Ehsan 47500, Malaysia; 19117928@imail.sunway.edu.my (H.-Y.C.); ronaldt@sunway.edu.my (S.-Y.T.); 19117936@imail.sunway.edu.my (M.L.-K.S.); 3Institute of Bio Science, University Putra Malaysia, Serdang 43400, Malaysia; kia.pooneh@gmail.com

**Keywords:** cellulose, chitosan nanoparticles, bionanocomposites, 5-fluorouracil, in vitro drug release, cytotoxicity assay, colorectal cancer

## Abstract

Cellulose and chitosan with remarkable biocompatibility and sophisticated physiochemical characteristics can be a new dawn to the advanced drug nano-carriers in cancer treatment. This study aims to synthesize layer-by-layer bionanocomposites from chitosan and rice straw cellulose encapsulated 5-Fluorouracil (CS-CF/5FU BNCs) using the ionic gelation method and the sodium tripolyphosphate (TPP) cross-linker. Data from X-ray and Fourier-transform infrared spectroscopy showed successful preparation of CS-CF/5FU BNCs. Based on images of scanning electron microscopy, 48.73 ± 1.52 nm was estimated for an average size of the bionanocomposites as spherical chitosan nanoparticles mostly coated rod-shaped cellulose reinforcement. 5-Fluorouracil indicated an increase in thermal stability after its encapsulation in the bionanocomposites. The drug encapsulation efficiency was found to be 86 ± 2.75%. CS-CF/5FU BNCs triggered higher drug release in a media simulating the colorectal fluid with pH 7.4 (76.82 ± 1.29%) than the gastric fluid with pH 1.2 (42.37 ± 0.43%). In in vitro cytotoxicity assays, cellulose fibers, chitosan nanoparticles and the bionanocomposites indicated biocompatibility towards CCD112 normal cells. Most promisingly, CS-CF/5FU BNCs at 250 µg/mL concentration eliminated 56.42 ± 0.41% of HCT116 cancer cells and only 8.16 ± 2.11% of CCD112 normal cells. Therefore, this study demonstrates that CS-CF/5FU BNCs can be considered as an eco-friendly and innovative nanodrug candidate for potential colorectal cancer treatment.

## 1. Introduction

Currently, conversion of natural-based residues into useful and novel materials may tackle financial and environmental issues [1]. In this regard, rice straw is an abundant lignocellulosic residue, which contains a high ratio of cellulose to use in a myriad of research fields [2,3]. As a nature gifted material, cellulose fiber is an important component of the wood cell wall and the most abundant organic polymer on earth [4]. Plant cellulose possesses advantageous and unique mechanical, optical and rheological characteristics, along with a sensitivity to the particular molecular structure of the antigen and pH-sensitivity for synthesis of novel polymeric nanodrug formulations [4]. The cellulose, however, does possess some drawbacks, including poor crease resistance and low solubility in solvent fluids. Its physiochemical characteristics suitably can be modified by esterification, etherification, de-polymerization, radical grafting and alkali treatments to impart exogenous groups over the cellulose structure without damaging its advantageous intrinsic characteristics [5].

Chitosan is the second most popular biopolymer after cellulose, with production of over 100 million tons per year [6]. It may be derived from chitin and is a cationic linear and natural amino-polysaccharide containing -(1-4)-linked d-glucosamine and *N*-acetyld-glucosamine in deacetylated and acetylated form, respectively [7]. Among diverse methods to synthesize layer-by-layer chitosan-based composites and nanoparticles (NPs), ionic gelation approach is an organic solvent-free solution, straightforward and a facile method with minimal toxicity [8]. In this method, the phosphate groups of sodium tripolyphosphate (TPP) may act as a physical crosslinking agent, which has advantages over emulsifying and chemical crosslinking agents, such as less toxicity to the organs and no destruction to the structure of the loaded-drugs in chitosan nanoparticles [9]. Polymer blends are increasingly important to fabricate innovative composites for various applications [10]. The blend of degradable polymers can merge the desirable properties of the polymers [11]. In addition, the crosslinking procedure might considerably improve physiochemical properties of the polymer composites [12]. In medically-related applications, the most popular antimicrobial coating agent on cellulose is currently chitosan to synthesize composites with suitable biocompatibility and water-rich structures to encapsulate both hydrophilic and hydrophobic drugs [13]. Furthermore, the bionanocomposites or nanohybrids of chitosan and cellulose possess intermolecular interactions, owing to H-bonds and Van der Waals forces [14]. Most importantly, chitosan-cellulose bionanocomposites or nanohybrids may possess a tremendous swelling capacity and water absorption ability to release the drug at the targeted cells [15]. Therefore, the biocompatibility and physiochemical properties of both cellulose and chitosan can be modified by using cellulose as a reinforcement and chitosan as a coating agent to construct a double polysaccharide composite biomaterial.

Colorectal cancer (CRC) is the third most diagnosed cancer with death of six lakhs per year [16]. 5-Fluorouracil (5FU) as an antimetabolite and fluorinated pyrimidine has been used for CRC therapy since three decades ago [17]. Albeit, the unwanted side-effects of using 5FU-based chemotherapy might cause diarrhea, stomatitis and gastrointestinal mucosal injury. The above side-effects can be abated by encapsulation of a minimal drug dosage into a biocompatible drug carrier. Of this, 5FU has been encapsulated in many drug carrier platform, such as, chitosan [18] and cellulose [19,20]. In this matter, some polysaccharides, including cellulose, chitosan, starch, xanthan gum, hyaluronic acid, and carrageenans can display pH/thermos-responsive properties to control the drug release dosage with effective anticancer actions [21].

Nano-sized medicines may favorably enhance circulation time, targeted activities, and safe drug delivery to decrease the issues of side effects from the chemotherapy treatments [22,23,24]. For this aim, biopolymer-based NPs may ideally avoid the immune system to increase circulation time of the NPs in blood to enhance the anticancer actions [25]. The drug delivery systems may affect the microenvironment of tumors, which is leaky and has a higher sensitivity to macromolecules compared to the normal cells [26]. Thus, drug encapsulation in natural polymers can be considered as an emerging research area for different pharmaceutical applications [24,25,27]. Furthermore, being impressively biodegradable and possessing an ability for the enzymatic degradation by colonic microbial agents, biopolymers (such as lignin-based NPs, glycogen and gelatin and cellulose fibers) may trigger probes for novel polymeric–anticancer nanodrug conjugates with advanced antiproliferative actions [24,25,27,28]. 

For example, 5FU-loaded composites of nanocellulose (obtained from commercial α-cellulose), chitosan and sodium alginate eliminated HT29 CRC cells, however, it unfavorably killed the normal HEK293 cells at concentrations between 50 to 100 µg/mL [14]. In a separate experiment, curcumin loaded nanocellulose caused three times higher effect compared to curcumin alone against HT29 CRC cells, albeit, the biocompatibility of the sample was not studied [29]. In recent investigations, the release of 5FU was controlled through nanocellulose/gelatin composites [30] and also 3D printed composites of cellulose nanofibrils and CaCO_3_ [31]. Both studies did not examine the anticancer action of the samples.

In this present study, cellulose fibers were extracted from rice straw waste. In addition, chitosan NPs and bionanocomposites of chitosan-cellulose were fabricated by the ionic gelation method using TPP as a cross-linker. Above all, 5FU was encapsulated in the bionanocomposites of chitosan as a coating agent and cellulose as a reinforcement for CRC analysis. To the best of our knowledge, this is the first time that rice straw cellulose was coated with chitosan for anticancer drug 5FU carrier in potential CRC treatments. The physiochemical properties of the synthesized samples were evaluated by X-ray powder diffraction (XRD), scanning electron microscopy (SEM), energy dispersion X-ray spectroscopy (EDX), thermogravimetric analysis (TGA), dynamic light scattering (DLS), Fourier-transform infrared spectroscopy (FTIR), and the swelling analysis. The drug absorbance and release of 5FU-loaded in the bionanocomposites were evaluated by using ultraviolet–visible (UV) spectroscopy. In vitro cytotoxicity assays evaluated the biocompatibility and anticancer activity of the fabricated samples against CCD112 colon normal and HCT116 CRC cell lines.

## 2. Materials and Methods

### 2.1. Materials

Rice straw waste was obtained from Malaysian Agricultural Research and Development Institute (MARDI), Selangor, Malaysia. Potassium hydroxide (KOH, 85%), sodium chlorite (NaClO_2_, 80%), acetic acid glacial (CH_3_COOH) (98%), chitosan (low molecular weight, 190,000–310,000 degree of acetylation), Tween-80 and TPP were all purchased from Sigma Aldrich (St. Louis, MO, USA). 5FU, 99%, 5-Fluoro-2,4(1H,3H)-pyrimidinedione (ACD CODE MFC D00006018) with a molecular weight of 130.08 g/mol was purchased from ACROS ORGANICS part of Thermo Fisher Scientific, Branchburg, NJ, USA. The chemicals were used without further purification. All glassware used was washed with distilled water and dried before used.

### 2.2. Synthesis 

#### 2.2.1. Extraction of Cellulose Fibers from Rice Straw Waste

CF was extracted from rice straw waste through a series of chemical treatments. The rice straw powder (30 g) was dewaxed by using a soxhlet instrument as it contained a 450 mL solution of toluene/ethanol 2:1 (*v*/*v*) for 12 h at 70 °C. After that, the sample was washed with distilled water three times and mixed with a sodium chloride solution (1.4%), in which dropwise adding acetic acid adjusted the pH to around 4 at 70 °C under 100 rpm magnetic stirring for 5 h. The sample was washed three times and then treated with KOH solution (5%) for 12 h, followed by pouring 10-fold ice cubes into the sample solution to obtain cellulose. The cellulose solution was washed and centrifuged (Tabletop Centrifuge Kubota, Model: 2420) three times at 4000 rpm for 8 min. Finally, the suspension was freeze-dried (FreeZone 1.0 L Benchtop Freeze Dry System, Kansas City, MO, USA) at −45 °C for 48 h and termed as CF, which stored at −4 °C for further analysis. 

#### 2.2.2. Synthesis of Chitosan-Cellulose Fiber Bionanocomposites to Encapsulate 5-Fluorouracil 

Ionic gelation method was performed to synthesize layer-by-layer of three chitosan-based samples: (i) chitosan-cellulose fibers bionanocomposites encapsulated 5FU (CS-CF/5FU BNCs), (ii) chitosan-cellulose fibers bionanocomposites (CS-CF BNCs), and (iii) chitosan nanoparticles (CS NPs). First, three 250 mL beakers respectively contained 80 mL mixture solution of 1.0% acetic acid and 0.250 g of chitosan powder (low molecular weight). Then, 2% (*v*/*v*) of Tween-80 as a stabilizer was respectively added to each solution and mixed gently for 45 min to obtain the three chitosan solutions. To prepare CS-CF/5FU BNCs, 0.125 g CF and 0.01 g 5FU were mixed with one of the prepared chitosan solutions and then homogenized vigorously (DAIHAN-brand Homogenizer, Wonju-Si, Gang-Won -Do, Korea) at 9000 rpm for 7 min. After that, 0.50 g of TPP cross-linker was dissolved in 15 mL deionized water and added dropwise to the CS-CF/5FU solution under the continuously vigorous stirring of the homogenizer for another 45 min. The mixture solution was washed with distilled water and centrifuged three times at 2500 rpm for 7 min at 25 °C. Finally, the sample was freeze-dried for 16 h and stored at −4 °C for further analysis. 

To prepare CS-CF BNCs, the above procedure was performed, however, without adding 5FU to the CS-CF solution.

To prepare CS NPs, the above method was used, however, without adding CF and 5FU to the CS solution.

### 2.3. Characterization

#### 2.3.1. Physicochemical Analysis

XRD (Philips, X’pert, Cu Ka, Amsterdam, North Holland, The Netherlands) at an ambient condition was used to evaluate the structure of the samples. The sample was compressed between two smooth glass films and the XRD analysis was carried out in dispersion 2 angles of 5°– 80° at a step size of 0.02° with 2 s/step as scanning rate using a voltage of 45 kV, a Ni-filtered Cu K radiation (=1.5406 A)° and a filament current of 40 mA. The SEM images were taken via using an Electro-Scan SEM instrument (model JSM 7600 F SEM, Tokyo, Kantō, Japan) attached to EDX to study the elemental composition of the sample. A low-acceleration voltage (10 kV) was used to prevent the degradation of the sample. Thermal analysis was carried out though using TGA (STA F3 Jupiter, Selb, Bavaria, Germany) Q50 V20 at a heating rate of 10 °C/min under a nitrogen atmosphere (10 mL/min) from 10 °C to 800 °C. For DLS analysis, an Anton Paar instrument (Litesizer 500, Graz, Styria, Austria) was used to measure the hydrodynamic size of the synthesized samples in buffer solutions (100 μg/mL) at various pH values of 1.2, 7.4 and 12 at 37 °C. Hydrochloric acid (HCl) and sodium hydroxide (NaOH) were used to adjust the pH 1.2 and 12, respectively. FTIR spectroscopy (ThermoNicolet, Waltham, MA, USA) determined the chemical and super-molecular structural analysis of the samples under an ambient condition. First, crushing and mixing of the sample with KBr at a ratio of 1:100 *w*/*w* to prepare a transparent pellet and the spectra of the plate was evaluated under a transmittance mode in a range between 4000 cm^−1^ to 400 cm^−1^ with a 4 cm^−1^ resolution and an accumulation of 128 scans. The swelling ratio of the sample was measured. For this aim, the sample was immersed in the solutions with various pH values of 1.2, 7.4, and 12 at 37 °C under 75 rpm magnetic stirring [32]. At each interval, the sample was collected from the solution and blotted on a filter paper to eliminate excess water and it was immediately weighed to determine the weight of the wet sample. The swelling ratio of the sample was measured as W_t_/W_0_, in which W_t_ and W_0_ are the obtained wet weights of the sample at an arbitrary and initial time, respectively. For the DLS analysis and swelling experiment, individual tests were repeated three times, which the data were indicated as mean ± standard deviation for all triplicates within an independent test.

#### 2.3.2. Encapsulation Efficiency Study of 5-Fluorouracil

UV–vis spectrophotometry (UV-1600, Shimadzu, Kyoto, Kansai, Japan) provided a calibration curve at λ_max_ = 266 nm from known concentrations of 5FU (5–25 ppm). This followed by calculating the amount of the drug-loaded into CS-CF/5FU BNCs by using Equation (1) to obtain drug encapsulation efficiency% [33].
(1)$$\mathrm{Encapsulation}\;\mathrm{efficiency}\;\left(\%\right)=\frac{\;\left(\mathrm{Initial}\;\mathrm{drug}\;\mathrm{amount}\;\mathrm{in}\;\mathrm{formulation}\;\left(\mathrm{mg}\right)-\;\mathrm{Unentrapped}\;\mathrm{drug}\;\left(\mathrm{mg}\right)\right)}{\mathrm{Initial}\;\mathrm{drug}\;\mathrm{amount}\;\mathrm{in}\;\mathrm{formulation}\;\left(\mathrm{mg}\right)}\times 100$$

#### 2.3.3. A Comparative Study of In Vitro Release of 5-Fluorouracil Drug from CS-CF/5FU BNCs

For the drug release performance of CS-CF/5FU BNCs, a 5 mL dialysis bag (molecular weight cut-off between 12,000 and 14,000 Da) was used according to reported studies with slight modification [20,34]. Before the experiment, the bag was soaked for 12 h in the release media of the simulated colorectal fluid (phosphate-buffered saline (PBS) at pH 7.4). Then, the dialysis bag with the two ends tied as contained the solution mixture of 5 mg of CS-CF/5FU BNCs and 2 mL of the release media. The bag was completely immersed into a 40 mL of the release media maintained under constant stirring of 100 rpm at 37 °C in the stoppered bottle. A 1 mL aliquot was withdrawn from the system at the selected time and characterized by UV–vis spectrophotometry at λ_max_ = 266 nm. The same study was performed in HCl buffer solution at pH 1.2 for the simulated release of the 5-FU drug in the gastric fluid without enzymes. The following Equation (2) calculated and compared the drug release results from the fluids with the two different pH values:(2)$$\mathrm{Drug}\;\mathrm{release}\;\left(\%\right)=\frac{\mathrm{Released}\;\mathrm{drug}\;\mathrm{at}\;\mathrm{time}\;‘t’\;}{\mathrm{Total}\;\mathrm{drug}\;\mathrm{in}\;\mathrm{the}\;\mathrm{sample}\;\left(\mathrm{CS}-\mathrm{CF}/5\mathrm{FU}\;\mathrm{BNCs}\right)}\times 100\;$$

#### 2.3.4. Cell Lines and Reagents

Human HCT116 colorectal cancer (CRC) (ATCC CCL-247) and CCD112 normal colon (ATCC CRL-1541) cell lines were purchased from ATCC (VA, USA) and cultured according to ATCC’s recommendation [35,36]. The cell lines were maintained in Dulbecco’s Modified Eagle’s medium (DMEM) supplemented with 10% fetal bovine serum (FBS) (Gibco) and 1% penicillin/streptomycin (Gibco) under standard cell culture conditions.

#### 2.3.5. In Vitro Cytotoxicity Assay

In vitro cytotoxicity assays were performed using CellTiter 96 Aqueous One Solution kit (MTS reagent) (#G3582, Promega), according to the manufacturer’s instruction with slight modification as previously described [37]. Briefly, 5000 HCT116 and CCD112 cells per well (100 µL/well) were separately seeded onto a 96 well plate and incubated overnight at 37 °C in a 95% humidified incubator with 5% CO_2_. The next day, 2-fold serially diluted samples at concentrations of 0, 7.81, 15.62, 31.25, 62.53, 125, 250 and 500 (100 µL/well) were added into the wells and the plate was incubated for 72 h at 37 °C in the 5% CO_2_ incubator. Then, 20 µL of the MTS reagent per well was added into the plate and incubated for an additional 3 h at 37 °C in the 5% CO_2_ incubator. The optical density (OD) was then measured at 490 nm using a multimode microplate reader (Tecan). The dose–response graph was plotted by calculating the percent cell viability using Equation (3) [38]:(3)$$\%\;\mathrm{Cell}\;\mathrm{viability}=\frac{\mathrm{OD}\;\mathrm{of}\;\mathrm{sample}\;\mathrm{well}\;(\mathrm{mean})}{\mathrm{OD}\;\mathrm{of}\;\mathrm{control}\;\mathrm{well}\;(\mathrm{mean})}\times \;100\;$$

The inhibitory concentration causing 50% growth inhibition (IC_50_) was determined through an online calculator (https://www.aatbio.com/tools/ic50-calculator (accessed on 5 May 2021)) as previously described [35]. 

#### 2.3.6. Statistical Analysis

Independent experiments were performed three times and data were expressed as mean ± standard deviation for all triplicates within an individual experiment. Data were analyzed with a Student’s *t* test using SPSS version 26.0. *p* < 0.05 was considered significant.

## 3. Results and Discussion

Scheme 1 depicts the synthesis of a novel nanocomposites of rice straw CF and CS encapsulated 5FU for the in vitro CRC analysis. CF was successfully extracted from rice straw waste by a series of chemical modifications, such as, bleaching, delignification, and alkali treatments. Sodium chloride solution and potassium hydroxide degraded lignin and hemicelluloses of the rice straw waste, respectively [39]. Then, the delignificated rice straw waste was treated with 5% potassium hydroxide to isolate CF containing nano-scale diameter and almost a uniform structure. This method was used by different studies, which found similar results from the extracted cellulose of the newspaper waste [40] and rice straw [41]. In a separate investigation [42], an increase in the ratio of potassium hydroxide from 5 to 15 weight% decreased the size of the abaca fibers; whereas, the 10 and 15 weight% of the potassium hydroxide undesirably decreased the strength of fibers comprising a twisted structure. Thus, we used proper treatments of 5% potassium hydroxide solution to extract cellulose from the rice straw waste.

Incorporation of dual polysaccharides of CF and CS in a layer-by-layer complex encapsulated the anticancer drug 5FU by using the ionic gelation method, where TPP, Tween-80, and acetic acid served as a cross-linking agent, stabilizer and hydrogen ion formation, respectively. The ionic gelation method is facile and responsible for the increased degree of cross-linking between the components of the CS-based composites [43]. CF as a reinforcement can be compatible with CS as a coating agent to synthesize dual biocompatible composites [7]. The characteristics of CS based complexes might be enhanced by reinforcement of cellulose. Also, CS as a coating agent could possibly show inherently biocompatibility, whereas its physiochemical characteristics could be enhanced by CF as a matrix to modulate the drug release and improve anticancer effects [44].

The ratio of the ingredients in the polymer composites is an important concern. The ratio between the chitosan powder to TPP cross-linker was 1:2 (*v*/*v*). Similarly, different research indicated that this ratio was suitable in which a reaction yield (75.0%), drug loading content (16.7%), and encapsulation efficiency (66.7%) were optimized to obtain chitosan NPs with desirable physiochemical properties. The study also used 1.0% acetic acid and 2% (*v*/*v*) Tween-80 in the chitosan solution. For the CS-CF composite, we use CF and chitosan with the ratio of 1:2 to obtain the spherical nanocomposite with size below 50 nm, as indicated in the SEM images. In a different study, 2.0% (*w*/*v*) chitosan, 1.2% (*w*/*v*) carboxymethyl-cellulose and 1.0% (*w*/*v*) scleroglucan were used to synthesize a nanocomposite hydrogels [45]. Based on SEM images, this sample was not as a spherical composite. In addition, the chitosan did not show homogeneous coating structure on the carboxymethyl-cellulose. This could be possibly due to using higher ratio of the cellulose and scleroglucan than chitosan. Studies by Samy et al. (2020) have shown that the ratio between chitosan and cellulose was 1:1 and epichlorohydrin acted as a cross-linker [46]. This composite showed size above 100 nm, according to the SEM images. In our study, therefore, the successful fabrication of CS-CF composites with size below 50 nm could be owing to blending the proper ratio of CS coating agent and CF reinforcement under high-speed homogenizing (9000 rpm), whereas TPP desirably acted as a crosslinking agent. Further, the CS-CF composites may possess van der Waals interactions with the anticancer drug 5FU. For the CS-CF/5FU sample preparation, only 0.01 g 5FU was added in the CS-CF solution. It is worth to mention that using the low amount of drug in the nano-carrier system may cause high drug encapsulation efficiency and prolonged drug release to obtain effective anticancer action. Studies by Nguyen et al. (2017) showed that using the same amount of 5FU (0.01 g) in chitosan solution and ionic gelation method for the improved drug encapsulation and drug release [47]. Also, studies by Hosokawa et al. (1991) also showed that encapsulated the same amount of 5FU (0.01 g) with poly (ethylene glycol) methyl ether-chitosan nano-gels [48]. Therefore, the ratio of the ingredients in CS-CF/5FU could be suitable to obtain a nanodrug complex with relevant physiochemical properties and anticancer effects.

Scheme 2 indicates the possible intermolecular chemical interactions in the bionanocomposites. The presence of hydrogen and Van der Waals forces in CS-CF/5FU BNCs could trigger intermolecular interactions in the composites [14,19]. During the sample preparation, cationic groups of chitosan (${\mathrm{NH}}_{3}^{+}$) could react with the ‒OH^−^ and phosphoric ions in the TPP solution to support the deprotonation of CS. As reported [49], the formation of CS cross-linked CF could be obtained from Schiff base reaction between primary amino and carbonyl groups of CS and CF, respectively. Furthermore, 5FU was encapsulated and conjugated into the composites potentially by hydrogen bonding and van der Waals interactions [7]. The freeze-drying process was performed for the synthesized samples. It was reported that the freeze-drying process might trigger a gentle vaporization of the bound water and also sublimation of the ice crystals upon the samples to increase the presence of –OH groups and produce white fluffy fibrous materials [50].

### 3.1. Physicochemical Characterization of CS-CF/5FU BNCs Using X-ray Powder Diffraction

XRD was used to evaluate the crystallographic structure of the fabricated samples. Figure 1a–e show the XRD results of rice straw waste, CF, CS NPs, CS-CF BNCs, and CS-CF/5FU BNCs, respectively. The samples containing CF exhibited a similar XRD pattern, representing that the chemical treatments did not destroy the cellulose structure. The diffraction peaks of CF was approximately at 2θ = 15.51°, 22.23° and 35.32°, similar to the normal cellulose-I structure [39]. The main crystalline region was at 22.23° with a strong intensity, presenting the proper crystallinity of CF. Therefore, the delignification, bleaching as well as alkali treatments on the rice straw waste effectively degraded the amorphous regions and liberated the crystal regions. From Figure 1c, CS NPs presented no peaks of crystallinity since the crystalline structure of CS was eliminated after its crosslinking with TPP [51]. CS-CF BNCs and CS-CF/5FU BNCs performed a similar pattern attributed to both CF and CS peaks approximately at 2θ = 22.23°, 15.78° and 9.04°, which is in a good agreement with the JCPDS card no. 04-0784 [52]. Noticeably, the peak at 15.78° is an overlapping peak between CF and CS. Compared to CF, the CS-CF/5FU BNCs sample showed lower crystallinity peaks, due to presence of the TPP cross-linker to abate the crystallinity [51]. Further results indicated that CS-CF/5FU BNCs did not have any sharp peak related to 5FU, since the drug was possibly entrapped within the carrier composites [53]. The 5FU alone could display a crystalline structure with a sharp diffraction peak at around 2θ = 28.71° [53]. Moreover, the crystalline structure of the drug alone could possibly become a non-crystalline after its encapsulation within the carrier [54]. The XRD results indicated that CS-CF/5FU BNCs possessed a structure as polysaccharides to entrap 5FU.

### 3.2. Physicochemical Characterization of CS-CF/5FU BNCs Using Scanning Electron Microscopy and Energy Dispersive X-ray Analyses

SEM provides information about surface topography of the samples. Figure 2a–d show SEM images and EDX results for CF, CS NPs and CS-CF/5FU BNCs, respectively. The extracted CF was mostly in a rod-shaped structure comprised of individual and organized nano-fibrils with an average width of 50.81 ± 3.6 nm. The size of the rice straw waste gradually decreased after degradation of the hemicellulose and lignin through dewaxing, delignification and alkali treatments to liberate CF with nearly a uniform structure. CS NPs presented spherical shapes (24.17 ± 1.1 nm) and CS-CF/5FU BNCs demonstrated that CS NPs coated the CF network. An average diameter of CS-CF/5FU BNCs was found to be 48.73 ± 1.5 nm. The size of CS NPs is related to the ratio between the TPP and the chitosan powder. According to a separate report [55], the chitosan and TPP with a ratio around 1:2 could cause the formation of CS NPs with appropriate properties and nano-scale dimension, therefore, it was used in this current study. The use of the layer-by-layer synthesis potentially caused that CF was coated with CS and it was almost homogenously distributed in the multi-layered particles of the BNCs with low agglomeration. This was similarly found in different investigations [44,56]. It can be possibly noticed from two images of the composites that most of the rod-shaped CF was entangled in the spherical CS particles; thus, CS-CF/5FU BNCs mainly indicated the spherical shape. Whereas, a few CF assembled clusters of CS particles. The CF structure as a reinforcement could potentially trigger a positive influence to host 5FU and increase the drug conjugation and absorbance within the composite; moreover, the CS coating on CF possibly enhanced the drug entrapment and encapsulation in the composites.

It can be understood from the EDX results of CF that the chemical treatments on rice straw waste dissolved the silica in aqueous ions to be subsequently replaced by carbon. Compared to the EDX results of CF, the composites of CS-CF showed increasing and decreasing carbon and oxygen content, respectively (Figure 2a–c). A separate study stated that decreasing the amount of oxygen in CS-CF composites may increase the biodegradability of the CS-based composites [2]. As shown in EDX layered image (Figure 2d), CS-CF/5FU BNCs showed elements related to CS and CF as well as a negligible ratio of the fluorine (F) from 5FU. The above SEM and EDX results indicated that the nanocomposites successfully were synthesized to entrap anticancer drug 5FU.

### 3.3. Physicochemical Characterization of CS-CF/5FU BNCs Using Dynamic Light Scattering

DLS was used to estimate the size distribution of the particles in the solution. Figure 3a–c show the DLS results with values for hydrodynamic size of the synthesized samples in distilled water. The suspensions of CF, CS NPs, and CS-CF/5FU BNCs showed the hydrodynamic size of 162.58 ± 4.31, 133.18 ± 3.46, and 197.56 ± 4.12 nm, respectively. The obtained size was larger in DLS compared to that from SEM. This can be explained that DLS indicates the size of the particle and the surrounding diffuse layer of the particle, however, the SEM size is attributed to the particle itself. For the colloidal stability in the aqueous media, the presence of particle–particle interactions affected the hydrophobic attraction energy between the particles to attract each other.

The dimensional effect of the prepared nanofluid samples under various conditions was demonstrated. In order to deliver nanodrug complex into the target cells, the pH sensitivity of the fabricated samples is a vital concern. As depicted in Table 1, effects of pHs on hydrodynamic size of the nanofluid sample were evaluated using the samples at the solution with different pH values of 1.2 (strong acid), 7.4 (neutral) and 12 (strong base). All the synthesized samples indicated the size below 220 nm in the solution at the harsh conditions of strong acidic (pH 1.2) and basic (pH 12). This may possibly show that the harsh conditions did not damage the nano-dimensional of the examined samples. The results indicated that an increase in the pH of the CF solution from 1.2 to 12 slightly increased the hydrodynamic size of CF from 135.56 ± 2.87 nm to 203.17 ± 4.81 nm. Compared to CF, both CS-CF BNCs and CS-CF/5FU BNCs showed a higher enhancement of the size with increasing the pH value. This may address the pH-sensitive nature of chitosan as also stated in a different study [57]. At the solution with pH 1.2, CS NPs, CS-CF BNCs and CS-CF/5FU BNCs showed hydrodynamic size of 71.61 ± 5.54, 109.03 ± 4.12, and 112.51 ± 4.09 nm, respectively; whereas these samples indicated the size of 246.09 ± 5.10, 275.34 ± 4.59 and 274.23 ± 5.11 nm at the solution with pH 12. This increase in the hydrodynamic size of the samples could be attributed to the particle agglomeration. Particle aggregation might trigger reduced repulsive on the particles surface, due to the rise in the pH of the CF and CS solution [57]. It could be understood from the above results that the CS coating could enhance the pH sensitivity of the CF solid support to synthesize CS-CF/5FU BNCs as a pH-sensitive nanodrug formulation.

### 3.4. Physicochemical Characterization of CS-CF/5FU BNCs Using Thermal Analysis 

The thermal stability of the samples was determined by TGA. The results of TGA and differential thermogravimetric analysis (DTGA) of the fabricated samples are shown in Figure 4a–e. CF, CS NPs, CS-CF BNCs, CS-CF/5FU BNCs, and 5FU showed the main thermal degradation at 308.58, 236.54, 269.56, 298.10 and 284.60 °C with the final residue of 12.07, 14.53, 36.81, 32.50 and 0.91 weight % at 800 °C, respectively. During the thermal degradation, carbonyl and carboxyl groups could cause the reduction of the chain size and rapture the bonds of the polysaccharides of CS and CF [58]. It stated that the cellulosic materials may have a single-step degradation in nitrogen atmosphere, however, it is a two-step thermal degradation in air atmosphere [59]. In this manner, CF displayed a single-step degradation. CS-CF and CS-CF/5FU BNCs indicated a higher final residue % and thermal stability than CF and 5FU alone, showing the needs for the polymer blend of chitosan-cellulose to encapsulate anticancer drug in an advanced nano-carrier system.

So, for the percentage of the ingredient used for the preparation of CS-CF/5FU BNCs, it has been determined that approximately 64.91, 32.64, and 2.60 % of the constituent compounds are related to CS, CF and 5FU, respectively.

### 3.5. Physicochemical Characterization of CS-CF/5FU BNCs Using Fourier-Transform Infrared Spectroscopy

FTIR can identify changes in the chemical structure of the sample by generating an infrared absorption spectrum. The FTIR results of rice straw waste, CF, CS NPs, CS-CF BNCs, CS-CF/5FU BNCs and 5FU are indicated in Figure 5a–f, respectively. The bleaching procedure on rice straw waste potentially triggered the development of C-H aromatic hydrogen groups. From Figure 5b, the peaks at 1524 and 1750 cm^−1^ are for C–O–C bonds and bleaching of hemicellulose, respectively [41]. Both peaks at 760 and 491 cm^−1^ are attributed to elimination of silica (Si–O–Si stretching) [41]. Further results presented that the peaks at 3352, 2891 and 1100 cm^−1^ might determine the stretching vibrations of -OH groups, C-H stretching, and the cellulose network structure, respectively [3]. In the anomeric region (950–700 cm^−1^), CF indicated a minor peak at 887 cm^−1^, due to presence of the glycosidic –C_1_ –O –C_4_ deformation of the β-glycosidic bond [60]. This possibly indicated the efficient chemical treatments to isolate CF. The absorption peaks are different between CS NPs and CS-CF BNCs, showing contribution of OH or NH and changes in the sugar ring, Van der Waals forces, dipole moments and hydrogen bonds [61]. In CS-CF/5FU BNCs, the CO stretching vibration at 1654 cm^−1^ can be owed to a good interaction between the drug and its nano-carrier. Furthermore, an overlapping between OH and NH bonds of 5FU possibly demonstrated a band at 2750–3500 cm^−1^, comparable with another study [19]. It stated that the amine groups of CS with carbonyl groups of CF may form functional groups of imines with carbon-nitrogen double bond [62]. 

As indicated in Figure 5d, CS-CF/5FU present a peak at 3385 cm^−1^, because of –NH_2_ and –OH stretching groups. The interaction between ${\mathrm{NH}}_{3}^{+}$ groups of CS and phosphate groups of TPP led to presence of CONH_2_ and NH_2_ groups with peaks at 1639 and 1548 cm^−1^, respectively [63]. Whereas, in the spectra of CS-CF/5FU, these peaks shifted to 1654 and 1541 cm^−1^, respectively. The bands at 1411 and 1016 cm^−1^, respectively, presented –CH_2_ and P=O stretching vibrations from phosphate groups [63]. Furthermore, the primary amino group of CS reacted with the carbonyl groups of CF, whereas acetic acid dissolved CS to form hydrogen ion and affect the crosslinking structure from the Schiff base bond [64]. As stated that the primary amine group in CS could trigger the in situ gelation in the CS-based composites [65].

In addition, the development of the Schiff’s base probably reduced the aldehyde groups on the CF surface for bonding with CS. From the spectrum of 5FU, the C–F stretching vibration is probably at 1244 cm^−1^. This peak was shifted to 1249 cm^−1^ in the spectra of the CS-CF/5FU sample [19]. As reported [66], two protonated amide groups in the 5FU molecule may lead to a positive charge of 5FU and hydrogen bonds with carboxyl bonds, albeit, there were nearly ionized together to form a hydrogen bond. In turn, the 5FU molecules were possibly conjugated to the CS structure [66]. The FTIR spectrum of CS-CF/5FU BNCs indicated only a few peaks attributed to the drug, due to an efficient entrapping of the drug within the polysaccharide composites. 

### 3.6. Swelling Analysis of CS-CF/5FU BNCs 

The swelling analysis may determine the amount of the absorbed liquid by the polymer materials. The swelling properties of composites can act importantly in the drug loading and drug release performance. Figure 6a-d show the swelling kinetic of the fabricated samples in media at different pHs at 37 °C. The samples in the solutions at different pHs showed the main swelling ratio in the first 4 h. The increased swelling index ratio could be related to the natural property of the sample containing porous interconnected fibers [67]. The sample approximately showed its equilibrium state after 12 h, comparable with different studies [32,67]. With an increasing time and pH value, the swelling increased that CF, CS NPs, CS-CF BNCs and CS-CF/5FU BNCs indicated the maximum swelling ratio of 2.91 ± 0.11, 3.49 ± 0.09, 3.77 ± 0.07 and 3.85 ± 0.10 in media at pH 12, respectively, after 36 h. Similarly, in different studies, the pH of solution controlled the swelling performances of polymer complexes, including polyvinyl alcohol/chitosan/TiO_2_ nanofibers [68], chitosan cross-linked poly (acrylic acid) hydrogels [69] and hydrogel composites of carboxymethylstarch-g-poly (acrylic acid)/palygorskite/starch/sodium alginate [70]. As presented in Figure 6a–d, CS NPs and the CS-CF composites displayed higher swelling properties compared to CF. Therefore, the CS structure domain the swelling properties of the CS-CF composites. The increasing swelling properties of the tested samples were in line with the hydrodynamic size in the solution with various pH values. The above swelling results could indicate the necessity of using CS to coat the CF matrix for increasing pH sensitivity of the CS-CF composites.

### 3.7. Drug Loading and Encapsulation Efficiency Percentage of CS-CF/5FU BNCs

Encapsulation efficiency was used to estimate the successful drug loading ratio in a nano-carrier system. Figure 7a(i–iii) indicate the 5FU absorbance, calibration curve at λ_max_ = 266 nm from known concentrations of 5FU (5–25 ppm), and UV-vis spectra of the unentrapped 5FU, respectively. From the UV absorbance and Equation (1) the encapsulation efficiency value was estimated to be 86 ± 2.75%. The alkali treated CF possibly possessed open bonds and good swelling properties to host 5FU [71]. The CS-CF composites could show the OH groups for effectively binding and releasing 5FU by ionic interactions, as reported [72]. Furthermore, 5FU is a heterocyclic aromatic organic compound with a low molecular weight to be diffused well within open pores and substrate of the CF [73]; whereas the CS coated both CF and 5FU to improve encapsulation efficiency. From the value of encapsulation efficiency, the synthesized composites could be a suitable nano-carrier for the 5FU encapsulation in cancer treatment.

### 3.8. In Vitro Drug Release of CS-CF/5FU BNCs

In vitro drug release analysis was used to determine the release kinetics of the drug-loaded samples. Figure 7b,c shows UV-vis spectra and the release kinetics of 5FU from CS-CF/5FU BNCs, respectively. As seen in Figure 7c, the time taken was 8 h for the drug release of 50.60 ± 1.88% and 28.04 ± 1.14% at the release fluids with pH 7.4 and pH 1.2, respectively. After 36 h, the total 5FU release was lower in the simulated acidic fluid at pH 1.2 (42.37 ± 0.43%) compared to that in the simulated colorectal fluid at pH 7.4 (76.82 ± 1.29%). This can be explained that CS-CF/5FU BNCs became smaller in the acidic media [67]. In the human body, the above may possibly result in a safe transferring of CS-CF/5FU BNCs through the acidic fluid of the stomach and then it may begin to break up in the colorectal fluid [67]. The low molecular weight of 5FU may cause a tendency for consecutive release, albeit, the cellulosic material as an amphipathic drug carrier and even excipient may modulate the release dosage [73,74]. The drug release showed an analogous trend with the swelling ratio of the samples, as schematically depicted in Figure 7d. An investigation stated that the controlled release ratio could be related to the interaction between 5FU and –COOH functionalities presented in the cellulosic material as it decreased when the swelling capacity increased [19]. The swelling may lead to polymer relaxation and gelling of 5FU to obtain a controlled drug release profile from CS-CF/5FU BNCs [74]. CS-CF/5FU BNCs presented a sustained and pH-sensitive release behavior, therefore, it may be considered as an attractive drug formulation candidate for modern CRC treatments.

### 3.9. In Vitro Cytotoxicity Assays

The biocompatibility and potential anticancer effects of the samples were studied by in vitro cytotoxicity assays and antibacterial assay (Appendix A Appendix A). The cytotoxicity of CF, CS NPs, CS-CF BNCs, CS-CF/5FU BNCs against HCT116 CRC and CCD112 normal colon cell lines are shown in Figure 8a–d. CF exhibited almost no prominent killing towards both normal and cancer cell lines (Figure 8a). Different reports stated that the wood-based nanocelluloses possess no adverse toxicity and based on ISO standard 10993-5, it does not have toxicity above 30% [75]. From Figure 8b, 15.62 µg/mL of CS NPs was enough to eliminate 26.46 ± 6.52% of the CRC cells without any toxicity on the CCD112 normal cell. In addition, the highest concentration (500 µg/mL) of CF, CS NPs, and CS-CF BNCs showed an appropriate biocompatibility without any damage against the normal cells. Among the samples without the loaded-5FU, CS-CF BNCs displayed better anticancer effect (41.62 ± 3.15%), showing a suitable contribution between CS and CF in the composites. As previously stated [15], the therapeutic potency of the chitosan-based samples could be explained that the positive charge of chitosan neutralized the negative charge on the tumor cell surface [14]. A different study also reported that the strong bonding between the primary amino group of chitosan and the carbonyl of cellulose could block bacteria from crossing the biocomposites [49]. It was indicated that the anticancer effects and permeation enhancement of CS-based materials could be related to the primary amine group in CS [65].

It can be noticed from Figure 8d that certain concentrations of CS-CF/5FU BNCs gave only negligible damage on the normal cells, while exhibiting effective anti-proliferation actions towards the cancer cells. Notably, CS-CF/5FU BNCs at 250 µg/mL concentration killed 56.42 ± 0.41% of the cancer cells and just 8.16 ± 2.11% of the normal cells. This anticancer effect might be attributed to the properties of the synthesized double polysaccharides to encapsulate a sufficient amount of 5FU for potentially improving mobility, binding ability, and colloidal stability to attach on the cell surface, as reported earlier [75]. Besides, based on a different study [17], polysaccharide complex might conjugate onto cancer cells to improve selectivity and decrease the drug leakage from the selected cells for saving the drug from unwanted degradation and elimination [16]. The polymeric network of CS-CF entrapped 5FU to potentially obtain a prolonged release dosage of the drug, similar to a separate report [16]. In the drug-loaded chitosan materials, the primary amine groups can possibly cause the controlled drug release [65]. As shown in Table 2, no activity was seen in the synthesized samples on the normal cells which suggests its high biocompatibility. Among all the synthesized samples, the IC_50_ happened only for the drug loaded sample that CS-CF/5FU BNCs showed the IC_50_ of 228.27 μg/mL on the cancer cells. It could be understood from the results of in vitro cytotoxicity assays that the polysaccharide samples of CF and CS have a suitable biocompatibility and CS-CF/5FU BNC suggests promising anticancer actions.

## 4. Conclusions

In this study, lignin and hemicellulose of rice straw waste were degraded by a series of chemical modifications include dewaxing, bleaching, and alkali treatments to liberate CF containing organized nanofibrils. By using the ionic gelation method, CS-CF composites encapsulated 5FU to synthesize nanodrug candidate of CS-CF/5FU BNCs for in vitro CRC analysis. The composites indicated the XRD and FTIR peaks related to the CS cross-linked CF. The SEM images of CS-CF/5FU BNCs showed that the spherical CS NPs covered the rod-shaped CF reinforcement. Based on swelling and hydrodynamic size analysis, the composites indicated a pH responsive behavior. Analogously, CS-CF/5FU BNCs exhibited a lower release ratio in media simulating the gastric fluid at pH 1.2 compared to the colorectal fluid at pH 7.4. In cytotoxicity assays, all the synthesized polysaccharide samples did not indicate the IC_50_ against the CCD112 normal colon cells, showing their proper biocompatibility and safety for anticancer applications. Among the fabricated samples, CS-CF/5FU BNCs only displayed the IC_50_ value (228.27 μg/mL) against the HCT116 cancer cells. Further results presented that the 250 µg/mL concentration of CS-CF/5FU BNCs showed better anticancer actions than other concentrations. The intriguing implication of this research is that using a very low dosage of 5-FU encapsulated in CS/CF composites could reduce the manufacturing cost and make it an effective green nanodrug candidate for anticancer actions against CRC.

## Data Availability

The data presented in this study are available on request from the corresponding author.

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
