# Peer review of "5-Fluorouracil Encapsulated Chitosan-Cellulose Fiber Bionanocomposites: Synthesis, Characterization and In Vitro Analysis towards Colorectal Cancer Cells"

_nanomaterials, 2021, doi:10.3390/nano11071691_

Round 1

Reviewer 1 Report

The authors prepared well the revised manuscripts. If the author can think about the following comments and add related contents, the completion of the paper will be higher.

  1. Please describe why chitosan and cellulose fiber were used "together" in your study.
  2. How was the mixing ratio of the ingredients determined?

Author Response

REVIWER 1:

Q1. Please describe why chitosan and cellulose fiber were used "together" in your study.

A1. The necessity of the chitosan-cellulose combination as a drug encapsulated biopolymers as well as its role in the gradual release are explained in the following addresses:

  • Page 6, Lines 259-262: The CS as a coating agent could possibly show inherently biocompatibility, whereas, its physiochemical characteristics could be enhanced by CF as a matrix to modulate the drug release and improve anticancer effects
  • Page 12, Lines 400-402: It could be understood from the above results that the CS coating could enhance the pH sensitivity of the CF solid support to synthesize CS-CF/5FU BNCs as a pH-sensitive nanodrug formulation
  • Page 14-15, Lines 484-489: As presented in Figure 6, CS NPs and the CS-CF composites displayed higher swelling properties compared to CF. Therefore, the CS structure domain the swelling properties of the CS-CF composites. The increasing swelling properties of the tested samples were in line with the hydrodynamic size in the solution with various pH values. The above swelling results could indicate the necessity of using CS to coat the CF matrix for increasing the pH sensitivity of the CS-CF composites.
  • Page 17, Lines 543-549: Among the samples without the loaded 5FU, CS-CF BNCs displayed better anticancer effect (41.62±3.15%), showing a suitable contribution between CS and CF in the composites for anticancer effects. As previously stated [15], the therapeutic potency of the chitosan-based samples could be explained that the positive charge of chitosan neutralized the negative charge on the tumor cell surface [14]. A different study also reported that the strong bonding between the primary amino group of chitosan and the carbonyl of cellulose could block bacteria from crossing the biocomposites [49].”
  • Page 15, Lines 502-504: Furthermore, 5FU is a heterocyclic aromatic organic compound with a low molecular weight to be diffused well within open pores and substrate of the CF. Whereas, the CS coated both CF and 5FU to improve encapsulation efficiency.

Q2. How was the mixing ratio of the ingredients determined?

A2. The following is mentioned in the manuscript to determine the mixing ratio of ingredients.

  • Figure 7 is shown the Thermal gravimetric analysis (TGA) for CF, CS NPs, CS-CF BNCs, CS-CF/5FU BNCs and 5FU samples. This method can be a suitable way for determined the percentage of ingredients used for synthesis of BNCs.
  • Page 12, Line 420-422: So, for the percentage of the ingredient used for the preparation of CS-CF/5FU BNCs, it has been determined that 64.91, 32.64, and 2.60% of the constituent compounds are related to CS, CF, and 5FU, respectively.

  • Page 6, lines 263-270: The ratio between the chitosan powders to TPP cross-linker was 1:2 (v/v). Similarly, a different research indicated that this ratio was suitable in which a reaction yield (75.0 %), drug loading content (16.7 %), and encapsulation efficiency (66.7%) were optimized to obtain chitosan NPs with desirable physiochemical properties. The study also used 1.0% acetic acid and 2% (v/v) Tween-80 in the chitosan solution. For the CS-CF composite, we use CF and chitosan with the ratio of 1:2 to obtain the spherical nanocomposite with size below 50 nm, as indicated in the SEM images.
  • A number of references to show the methods of measuring the ratio of ingredients:
  1. Page 6, Line 275-276: Studies by Samy et al. (2020) have shown that the ratio between chitosan and cellulose was 1:1 and epichlorohydrin acted as a cross-linker [46].
  2. Page 6, Line 285-287: Studies by Nguyen et al. (2017) shown that using the same amount of 5FU (0.01g) in chitosan solution and ionic gelation method for the improved drug encapsulation and drug release [47].
  3. Page 6, Line 287-289: Also, studies by Hosokawa et al. (1991) also shown that encapsulated the same amount of 5FU (0.01 g) with poly (ethylene glycol) methyl ether-chitosan nano-gels [48].

Scheme 1 on page 7 was modified:

Reviewer 2 Report

My suggestions have been fully addressed. 

Author Response

REVIWER 2:

My suggestions have been fully addressed.

Author Response: We highly appreciate from your positive view for our research paper.

Reviewer 3 Report

The manuscript by Yusefi and co-workers describe drug encapsulation and release with chitosan/cellulose polymer blends. The work is of interest to the readers of nanomaterials. There is a good amount of data presented in the manuscript and most of the discussions are sound. However, there are multiple shortcomings to be addressed before further consideration.

1) A detailed schematic of the preparation and encapsulation should be provided to enhance understanding of the work. The new concept and novelty of the work should be highlighted.

2) What is the basis for the possible intermolecular chemical interactions between active functional groups presented in the manuscript? The authors should perform relevant experiments to prove the role and existence of the assumed interactions. For instance NMR titration could provide useful information.

3) It is unclear how the errors were derived. Figure captions (e.g. Figures 6, 8) should have a short description on how many samples were analyzed and if independently prepared materials were used.

4) Drug encapsulation in natural materials is an emerging topic and recent diverse examples should be acknowledged in the introduction (10.1016/j.jcis.2020.11.072; 10.1016/j.biomaterials.2016.12.034; 10.3390/polym13040542).

5) The robustness of the prepared materials under harsh conditions should be demonstrated.

6) Chitosan often shows antibacterial activity. Due to the nature of the current application as drug delivery, the authors should test if the prepared material carriers have antibacterial activity.

7) The authors should justify the need for the polymer blend of chitosan-cellulose. Why a single polymer is not sufficient, i.e. either chitosan or cellulose? Polymer blends are increasingly important and they should be briefly introduced (10.1021/acsapm.0c00455; 10.1021/acsanm.8b01563; 10.1021/acsapm.9b00993).

8) The ratio of the ingredients in the composite material should be investigated and reported.

9) What was the rationale for the selected concentrations? Some justification and practical relevance should be mentioned during the discussions.

Author Response

The manuscript by Yusefi and co-workers describe drug encapsulation and release with chitosan/cellulose polymer blends. The work is of interest to the readers of nanomaterials.

There is a good amount of data presented in the manuscript and most of the discussions are sound. However, there are multiple shortcomings to be addressed before further consideration.

Q1. A detailed schematic of the preparation and encapsulation should be provided to enhance understanding of the work. The new concept and novelty of the work should be highlighted.

A1. We have revised the sentences below for the novelty of this research:

  • Page 1, Lines 29-31: Therefore, this study demonstrates that CS-CF/5FU BNCs can be considered as an eco-friendly and innovative nanodrug candidate for potential colorectal cancer treatment.
  • Page 3, Lines 108-109: To the best of our knowledge, this is the first time that rice straw cellulose was coated with chitosan for anticancer drug 5FU carrier in potential CRC treatment.
  • Page 6, Lines 239-240: Scheme 1 depicts the synthesis of a novel nanocomposites of rice straw CF and CS encapsulated 5FU for the in vitro CRC analysis.

We gave revised Scheme 1, in Page 7.

  • Page 18 Lines 597-600: The intriguing implication of this research is that using a very low dosage of 5-FU encapsulated in CS/CF composites could reduce the manufacturing cost and make it an effective green nanodrug candidate for anticancer actions against CRC.

Q2. What is the basis for the possible intermolecular chemical interactions between active functional groups presented in the manuscript? The authors should perform relevant experiments to prove the role and existence of the assumed interactions. For instance NMR titration could provide useful information.

A2.  We have revised the FTIR results and also the sentences below for chemical interactions between active functional groups of the samples. We also studied and explained the schematic molecular interaction of in the bionanocomposites.

  • Page 7, Lines 295-306: Scheme 2 indicates the possible intermolecular chemical interactions in the bionanocomposites. The presence of hydrogen and Van der Waals forces in CS-CF/5FU BNCs could trigger intermolecular interactions in the composites. During the sample preparation, cationic groups of chitosan () could react with the ‒OH and phosphoric ions in the TPP solution to support the deprotonation of CS. As reported, the formation of CS cross-linked CF could be obtained from Schiff base reaction between primary amino and carbonyl groups of CS and CF, respectively. Furthermore, 5FU was encapsulated and conjugated into the composites potentially by hydrogen bonding and van der Waals interactions. The freeze-drying process was performed for the synthesized samples. It was reported that the freeze-drying process might trigger a gentle vaporization of the bound water and also sublimation of the ice crystals upon the samples to increase the presence of –OH groups and produce white fluffy fibrous materials.
  • Page 8 Scheme 2:

Scheme 2. The possible intermolecular chemical interactions between active functional groups in CS, CF and 5FU

  • We have revised the FTIR results for CS-CF BNCs and CS-CF/5FU BNCs in Figure. 5.
  • Page 13 and 14, Lines 447-456: As indicated in Figures 5d, CS-CF/5FU present a peak at 3385 cm-1, because of –NH2 and –OH stretching groups. The interaction between groups of CS and phosphate groups of TPP led to presence of CONH2 and NH2 groups with peaks at 1639 and 1548 cm-1, respectively. Whereas, in the spectra of CS-CF/5FU, these peaks shifted to 1654 and 1541 cm-1, respectively. The bands at 1411 and 1016 cm-1, respectively, presented –CH2 and P=O stretching vibrations from phosphate groups. Furthermore, the primary amino group of CS reacted with the carbonyl groups of CF, whereas, acetic acid dissolved CS to form hydrogen ion and affect the crosslinking structure from the Schiff base bond. As stated that the primary amine group in CS could trigger the in situ gelation in the CS-based composites.
  • Page 14, Lines 457-465: In addition, the development of the Schiff's base probably reduced the aldehyde groups on the CF surface for bonding with CS. From the spectrum of 5FU, the C–F stretching vibration is probably at 1244 cm−1. This peak was shifted to 1249 cm−1 in the spectra of the CS-CF/5FU sample. As reported, two protonated amide groups in the 5FU molecule may lead to a positive charge of 5FU and hydrogen bonds with carboxyl bonds, albeit, there were nearly ionized together to form a hydrogen bond. In turn, the 5FU molecules were possibly conjugated to the CS structure. The FTIR spectrum of CS-CF/5FU BNCs indicated only a few peaks attributed to the drug, due to an efficient entrapping of the drug within the polysaccharide composites.

Q3. It is unclear how the errors were derived. Figure captions (e.g. Figures 6, 8) should have a short description on how many samples were analyzed and if independently prepared materials were used.

A3.  The error bars represent the standard deviations. The number of replicates and repeats of experiment have been added under the figure captions of Figure 6 and 8. We have added the sentences below for the error bars:

  • Page 5, Lines 234-237: [3.6. Statistical Analysis]

Independent experiments were performed three times and data were expressed as mean ± standard deviation for all triplicates within an individual experiment. Data were analysed with a Student’s t test using SPSS version 26.0. p < 0.05 was considered significant.

  • Page 4, Lines 186-188: For the DLS analysis and swelling experiment, individual tests were repeated three times, which the data were indicated as mean ± standard deviation for all triplicates within an independent test.
  • Page 12, Lines 404-406: Table 1. Hydrodynamic size of the synthesized samples in the solutions with various pH values. Data is expressed as the mean ± standard deviation for triplicates within an individual experiment.

Also in the caption Figures 6, 7, and 8.

Q4. Drug encapsulation in natural materials is an emerging topic and recent diverse examples should be acknowledged in the introduction. (10.1016/j.jcis.2020.11.072; 10.1016/j.biomaterials.2016.12.034; 10.3390/polym13040542).

A4. We have added the examples as Reference [24] for 10.1016/j.jcis.2020.11.072, in Page 2, Lines 83-85.

  • Page 2, Lines 83-85: To produce nano-sized medicines to favorably enhance circulation time and enable targeted activities, safe drug delivery and the prevention of side effects may now be possible. [22-24].

We have added the examples as Reference [27] for 10.1016/j.biomaterials.2016.12.034 in Page 2 Lines 90-94. 

  • Page 2 Lines 90-94: Furthermore, being impressively biodegradable and possessing an ability for the enzymatic degradation by colonic microbial agents, biopolymers (such as lignin based NPs, glycogen and gelatin, and cellulose fibers) may trigger probes for novel biomedical applications and polymeric–anticancer nanodrug conjugates with advanced antiproliferative actions [24, 25-28].

We have added the examples as Reference [25] for 10.3390/polym13040542 in Page 2 Lines 85-86.

  • Page 2 Lines 85-86: For this aim, polymeric nanoparticles may ideally avoid the immune system to increase circulation time of the nanoparticles in blood to enhance the antibacterial and anticancer properties [25].

Q5. The robustness of the prepared materials under harsh conditions should be demonstrated.

A5. We have added the effects of pHs on hydrodynamic size and swelling of all the fabricated sample at the solution with different pH values of 1.2 (strong acid), 7.4 (neutral) and 12 (strong base).

  • Page 4, lines 172-175: For DLS analysis, an Anton Paar instrument was used to measure the hydrodynamic size of the synthesized samples in buffer solutions (100 μg/mL) at various pH values of 1.2, 7.4, and 12 at 37 °C. Hydrochloric acid (HCl) and sodium hydroxide (NaOH) were used to adjust the pH 1.2 and 12, respectively.
  • Page 11-12, Lines 383-402: The dimensional effect of the prepared nanofluid samples under various conditions was demonstrated. In order to deliver nanodrug complex into the target cells, the pH sensitivity of the fabricated samples is a vital concern. As depicted in Table 1, effects of pHs on hydrodynamic size of the nanofluid sample were evaluated using the samples at the solution with different pH values of 1.2 (strong acid), 7.4 (neutral) and 12 (strong base). All the synthesized samples indicated the size below 220 nm in the solution at the harsh conditions of strong acidic (pH 1.2) and basic (pH 12). This may possibly show that the harsh conditions did not damage the nano-dimensional of the examined samples. The results indicated that an increase in the pH of the CF solution from 1.2 to 12 slightly increased the hydrodynamic size of CF from 135.56±2.87 nm to 203.17±4.81 nm. Compared to CF, both CS-CF BNCs and CS-CF/5FU BNCs showed a higher enhancement of the size with increasing the pH value. This may address the pH-sensitive nature of chitosan as also stated in a different study [57]. At the solution with pH 1.2, CS NPs, CS-CF BNCs, and CS-CF/5FU BNCs showed hydrodynamic size of 71.61±5.54, 109.03±4.12, and 112.51±4.09 nm, respectively. Whereas, these samples indicated the size of 246.09±5.10, 275.34±4.59, and 274.23±5.11 nm at the solution with pH 12. This increase in the hydrodynamic size of the samples could be attributed to the particle agglomeration. Particle aggregation might trigger reduced repulsive on the particles surface, due to the rise in the pH of the CF and CS solution [57]. It could be understood from the above results that the CS coating could enhance the pH sensitivity of the CF solid support to synthesize CS-CF/5FU BNCs as a pH-sensitive nanodrug formulation.

Table 1. Hydrodynamic size of the synthesized samples in the solutions with various pH values. Data is expressed as the mean ± standard deviation for triplicates within an individual experiment.

Hydrodynamic particle size (nm)

Sample

pH 1.2

pH 7.4

pH 12

CF

135.56±2.87

174.43±3.28

203.17±4.81

CS

71.61±5.54

140.09±4.70

246.09±5.10

CS-CF BNCs

109.03±4.12

198±3.25

275.34±4.59

CS-CF/5FU BNCs

112.51±4.09

203.52±2.94

274.29±5.11

  • Page 4, Line 180-182: The swelling ratio of the sample was measured. For this aim, the sample was immersed in buffer solutions with various pH values of 1.2, 7.4, and 12 at 37 °C under 75 rpm magnetic stirring [32].
  • Page 14-15, Line 477-489: With an increasing time and pH value, the swelling increased that CF, CS NPs, CS-CF BNCs, and CS-CF/5FU BNCs indicated the maximum swelling ratio of 2.91±0.11, 3.49±0.09, 3.77±0.07, and 3.85±0.10 in media at pH 12, respectively, after 36 h. Similarly, in different studies, the pH of solution controlled the swelling performances of polymer complexes, including polyvinyl alcohol/chitosan/TiO2 nanofibers [68], chitosan cross-linked poly(acrylic acid) hydrogels [69], and hydrogel composites of carboxymethylstarch-g-poly (acrylic acid)/palygorskite/starch/sodium alginate [70]. As presented in Figure 6, CS NPs and the CS-CF composites displayed higher swelling properties compared to CF. Therefore, the CS structure domain the swelling properties of the CS-CF composites. The increasing swelling properties of the tested samples were in line with the hydrodynamic size in the solution with various pH values. The above swelling results could indicate the necessity of using CS to coat the CF matrix for increasing pH sensitivity of the CS-CF composites.”

Figure 6. Swelling kinetic of CF, CS NPs, CS-CF BNCs, and CS-CF/5FU BNCs initiated from the wet state in two different media at pH 1.2, 7.4, and 12. Data is expressed as the mean ± standard deviation for triplicates within an individual experiment.

Q6. Chitosan often shows antibacterial activity. Due to the nature of the current application as drug delivery, the authors should test if the prepared material carriers have antibacterial activity.

A6. We have performed the antibacterial assay using the 4 samples against gram-positive Staphylococcus aureus and gram-negative Escherichia coli. However, no antibacterial action was seen. The data has been added into the supplementary data:

  1. Materials and Methods

2.3.6. Antibacterial Assay

A broth micro-dilution method was used to determine the minimum inhibitory concentration (MIC) of the samples using the Clinical and Laboratory Standards Institute (CLSI) protocols as described previously [35,36]. Gram-positive (Staphylococcus aureus ATCC 23235) and gram-negative (Escherichia coli ATCC 11775) bacterial strains were used in this study. Single colony of fresh bacterial culture (12–18 h) was isolated from Mueller Hinton agar (MHA) plates and inoculated into sterile Mueller Hinton broth (MHB). The culture was grown overnight (12–18 h) prior to the experiments. Next day, the bacterial concentration was standardized to an optical density (OD) of 600 nm (approximately 108 CFU/mL) with MHB. Two-fold serial dilutions of samples were prepared in 96-well plates to give final test concentrations of 0, 15.62, 31.25, 62.5, 125, 250, 500 and 1000 µg/mL per well. 10 µL of bacterial suspension equivalent to 106 CFU/mL of exponentially growing bacterial cells were added to the wells. The plates were incubated at 35 + 2 °C for 18 h. Following the overnight incubation, the plate was then read for the absorbance at 600 nm using microplate reader (Tecan) to determine the MIC values. Three independent experiments were performed and the data are expressed as the mean ± standard deviation for all triplicates within an individual experiment. 

  1. Results and Discussion

3.10. Antibacterial Assay

As CS, CS-based nanoparticles and nanocomposites have been known with their potent antibacterial action against a broad range of bacterial strains [1,2]. We tested the antibacterial potential of CS-containing compounds along with CF. However, no significant antibacterial activity of CF, CS NPs, CS-CF BNCs, and CS-CF/5-FU BNCs was seen in gram-positive Staphylococcus aureus and gram-negative Escherichia coli.

Figure S1. Antibacterial activity of CF, CS NPs, CS-CF BNCs and CS-CF/5-FU BNCs against (a) gram-positive Staphylococcus aureus and (b) gram-negative Escherichia coli.

Q7. The authors should justify the need for the polymer blend of chitosan-cellulose. Why a single polymer is not sufficient, i.e. either chitosan or cellulose? Polymer blends are increasingly important and they should be briefly introduced (10.1021/acsapm.0c00455; 10.1021/acsanm.8b01563; 10.1021/acsapm.9b00993).

A7. The need for the polymer blend of chitosan-cellulose was explained in:

  • Page 6, Lines 259-261: CS as a coating agent could possibly show inherently biocompatibility, whereas, its physiochemical characteristics were enhanced by CF as a matrix to modulate the drug release and improve anticancer effects [44].
  • Page 12, Lines 400-402: It could be understood from the above results that the CS coating agent could enhance the pH sensitivity of the CF solid support to synthesize CS-CF/5FU BNCs as a pH sensitive nanodrug formulation.
  • Page 12, Lines 417-419: CS-CF and CS-CF/5FU BNCs indicated a higher final residue % and thermal stability than CF and also 5FU alone, showing the needs for the polymer blend of chitosan-cellulose to encapsulate anticancer drug in an advanced nano-carrier system.
  • Page 14-15, Lines 484-489: As presented in Figure 6, CS NPs and the CS-CF composites displayed higher swelling properties compared to CF. Therefore, the CS structure domain the swelling properties of the CS-CF composites. The increasing swelling properties of the tested samples were in line with the hydrodynamic size in the solution with various pH values. The above swelling results could indicate the necessity of using CS to coat the CF matrix for increasing pH sensitivity of the CS-CF composites.
  • Page 17, Lines 543-550: Among the samples without the loaded 5FU, CS-CF BNCs displayed better anticancer effect (41.62±3.15%), showing a suitable contribution between CS and CF in the composites for anticancer effects. As previously stated [15], the therapeutic potency of the chitosan-based samples could be explained that the positive charge of chitosan neutralized the negative charge on the tumor cell surface [14]. A different study also reported that the strong bonding between the primary amino group of chitosan and the carbonyl of cellulose could block bacteria from crossing the biocomposites [49].
  • Page 15, Lines 502-506: Furthermore, 5FU is a heterocyclic aromatic organic compound with a low molecular weight to be diffused well within open pores and substrate of the CF [73]. Whereas, the CS coated both CF and 5FU to improve encapsulation efficiency.

We have added the sentences below for polymer blends are increasingly important and briefly introduced via using the references [10], [11], and [12].

  • Page 2, lines 58-62: Polymer blends are increasingly important to fabricate innovative composites for various applications [10]. The blend of degradable polymers can possibly merge the advantageous properties of the polymers [11]. In addition, the crosslinking procedure might considerably influence the enhancement of physiochemical properties of the polymer composites [12].

We have revised the sentences below in Conclusion section.

  • Page 18, lines 589-587: Based on swelling and hydrodynamic size analysis, the CS coating agent increased the pH sensitivity of the CF reinforcement to synthesize CS-CF/5FU BNCs as a pH-sensitive nanodrug formulation.

Q8. The ratio of the ingredients in the composite material should be investigated and reported.

A8. We have revised the sentences below for the ratio of the ingredients in the polymer composites.

  • Page 6, lines 263-291: The ratio of the ingredients in the polymer composites is an important concern. The ratio between the chitosan powders to TPP cross-linker was 1:2 (v/v). Similarly, a different research indicated that this ratio was suitable in which a reaction yield (75.0 %), drug loading content (16.7 %), and encapsulation efficiency (66.7%) were optimized to obtain chitosan NPs with desirable physiochemical properties. The study also used 1.0% acetic acid and 2% (v/v) Tween-80 in the chitosan solution. For the CS-CF composite, we use CF and chitosan with the ratio of 1:2 to obtain the spherical nanocomposite with size below 50 nm, as indicated in the SEM images. In a different study [45], 2.0% (w/v) chitosan, 1.2% (w/v) carboxymethyl-cellulose, and 1.0% (w/v) scleroglucan were used to synthesize a nanocomposite hydrogels. Based on SEM images, this sample was not as a spherical composite. In addition, the chitosan did not show homogeneous coating structure on the carboxymethyl-cellulose. This could be possibly due to using higher ratio of the cellulose and scleroglucan than chitosan. In a separate study [46], chitosan cross-linked cellulose was prepared, that the ratio between chitosan and cellulose was 1:1 and epichlorohydrin acted as a cross-linker. This composite showed size above 100 nm, according to the SEM images. In our study, therefore, the successful fabrication of CS-CF composites with size below 50 nm could be owing to blending the proper ratio of CS coating agent and CF reinforcement under high speed homogenizing (9000 rpm), whereas, TPP desirably acted as a crosslinking agent. Further, the CS-CF composites may possess van der Waals interactions with the anticancer drug 5FU. For the CS-CF/5FU sample preparation, only 0.01 g 5FU was added in the CS-CF solution. It is worth to mention that using the low amount of drug in the nano-carrier system may cause high drug encapsulation efficiency and prolonged drug release to obtain effective anticancer action. A recent published research [47] reported using the same amount of 5FU (0.01g) in chitosan solution and ionic gelation method for the improved drug encapsulation and drug release. Another study [48] also encapsulated the same amount of 5FU (0.01 g) with poly (ethylene glycol) methyl ether-chitosan nanogels. Therefore, the ratio of the ingredients in CS-CF/5FU could be suitable to obtain a nanodrug complex with relevant physiochemical properties and anticancer effects.

We have revised schem1 in Page 7:

Scheme 1. A schematic process of synthesis of CS-CF/5FU BNCs for in vitro cytotoxicity assays towards CRC cell line.

Q9. What was the rationale for the selected concentrations? Some justification and practical relevance should be mentioned during the discussions

A9. We have revised the sentences below for extraction of CF from rice straw waste;

  • Page 6, lines 242-251: Sodium chloride solution and potassium hydroxide degraded lignin and hemicelluloses of the rice straw waste, respectively [39]. Then, the delignificated rice straw waste was treated with 5% potassium hydroxide to isolate CF containing nano-scale diameter and almost a uniform structure. This method was used by different studies, which found similar results from the extracted cellulose of the newspaper waste [40] and rice straw [41]. In a separate investigation [42], an increase in the ratio of potassium hydroxide from 5 to 15 weight% decreased the size of the abaca fibers; whereas, 10 and 15 weight% of the treatments undesirably decreased the strength of fibers comprising a twisted structure. Thus, we used proper treatments to extract cellulose from the rice straw waste.

We have revised the sentences below for the ratio of the ingredients in the polymer composites.

  • Page 6, lines 263-292: The ratio of the ingredients in the polymer composites is an important concern. The ratio between the chitosan powders to TPP cross-linker was 1:2 (v/v). Similarly, a different research indicated that this ratio was suitable in which a reaction yield (75.0 %), drug loading content (16.7 %), and encapsulation efficiency (66.7%) were optimized to obtain chitosan NPs with desirable physiochemical properties. The study also used 1.0% acetic acid and 2% (v/v) Tween-80 in the chitosan solution. For the CS-CF composite, we use CF and chitosan with the ratio of 1:2 to obtain the spherical nanocomposite with size below 50 nm, as indicated in the SEM images. In a different study [45], 2.0% (w/v) chitosan, 1.2% (w/v) carboxymethyl-cellulose, and 1.0% (w/v) scleroglucan were used to synthesize a nanocomposite hydrogels. Based on SEM images, this sample was not as a spherical composite. In addition, the chitosan did not show homogeneous coating structure on the carboxymethyl-cellulose. This could be possibly due to using higher ratio of the cellulose and scleroglucan than chitosan. In a separate study [46], chitosan cross-linked cellulose was prepared, that the ratio between chitosan and cellulose was 1:1 and epichlorohydrin acted as a cross-linker. This composite showed size above 100 nm, according to the SEM images. In our study, therefore, the successful fabrication of CS-CF composites with size below 50 nm could be owing to blending the proper ratio of CS coating agent and CF reinforcement under high speed homogenizing (9000 rpm), whereas, TPP desirably acted as a crosslinking agent. Further, the CS-CF composites may possess van der Waals interactions with the anticancer drug 5FU. For the CS-CF/5FU sample preparation, only 0.01 g 5FU was added in the CS-CF solution. It is worth to mention that using the low amount of drug in the nano-carrier system may cause high drug encapsulation efficiency and prolonged drug release to obtain effective anticancer action. A recent published research [47] reported using the same amount of 5FU (0.01g) in chitosan solution and ionic gelation method for the improved drug encapsulation and drug release. Another study [48] also encapsulated the same amount of 5FU (0.01 g) with poly (ethylene glycol) methyl ether-chitosan nanogels. Therefore, the ratio of the ingredients in CS-CF/5FU could be suitable to obtain a nanodrug complex with relevant physiochemical properties and anticancer effects.
  • Page 9, lines 344-357: An average diameter of CS-CF/5FU BNCs was found to be 48.73±1.5 nm. The size of CS NPs is related to the ratio between the TPP and the chitosan powder. According to a separate report [55], the chitosan and TPP with a ratio around 1:2 could cause the formation of CS NPs with appropriate properties and nano-scale dimension, therefore, it was used in this current study. The use of the layer-by-layer synthesis potentially caused that CF was coated with CS and it was almost homogenously distributed in the multi-layered particles of the BNCs with low agglomeration. This was similarly found in different investigations [44,56]. It can be possibly noticed from two images of the composites that most of the rod-shaped CF was entangled in the spherical CS particles; thus, CS-CF/5FU BNCs mainly indicated the spherical shape. Whereas, a few CF assembled clusters of CS particles (Figure 2ci’). The CF structure as a reinforcement could potentially trigger a positive influence to host 5FU and increase the drug conjugation and absorbance within the composite; moreover, the CS coating on CF possibly enhanced the drug entrapment and encapsulation in the composites.

Round 2

Reviewer 3 Report

The authors have addressed the comments thoroughly, and improved the manuscript.